# Potential Use of *Hyssopus officinalis* and *Borago officinalis* as Curing Ingredients in Pork Meat Formulations

**DOI:** 10.3390/ani10122327

**Published:** 2020-12-07

**Authors:** Marzena Zając, Iwona Duda, Łukasz Skoczylas, Małgorzata Tabaszewska

**Affiliations:** 1Department of Animal Product Technology, Faculty of Food Technology, University of Agriculture in Cracow, 30-149 Cracow, Poland; iwona.duda@urk.edu.pl; 2Department of Plant Product Technology and Nutrition Hygiene, Faculty of Food Technology, University of Agriculture in Cracow, 30-149 Cracow, Poland; lukasz.skoczylas@urk.edu.pl (Ł.S.); malgorzata.tabaszewska@urk.edu.pl (M.T.)

**Keywords:** meat curing, borage, hyssop, curing alternative, polyphenols, colour, meat

## Abstract

**Simple Summary:**

Health risks associated with nitrites as curing agents have led consumers to search for products without those additives. Herbs have been used in medicine for many years and are usually positively perceived by consumers. Good-quality products with medicinal plants may be an alternative for those who try to avoid additives other than salt and spices. *Hyssopus officinalis* and *Borago officinalis* were tested for their potential to be used as colour forming and antioxidant agents. Both herbs were used in pork meat formulations along with nitrate reducing bacteria. A colour formation similar to a control product containing nitrite was noted in all the samples. Borage had a stronger antioxidant effect. Those additives can be used as an alternative to nitrite cured pork products.

**Abstract:**

The replacement of nitrites in pork meat products has been a studied issue for many years. Due to potential health threats associated with these additives, consumers tend to search for alternative meat curing methods. In this study, *Hyssopus officinalis* and *Borago officinalis* were tested for their potential to be used as colour-forming and antioxidant agents. Dry plant samples from various sources were tested for fat, protein, ash, polyphenol and nitrate content. There were significant differences between the herbs depending on source. Two control samples (containing curing salt and sodium chloride with nitrate reducing bacteria) and samples with herbs (hyssop, hyssop with nitrate reducing bacteria, borage, borage with nitrate reducing bacteria)—0.5% of the meat mass—were prepared and stored for 15 days. In the samples with herbs and bacterial cultures, a red colour was developed, the TBARS values were low and DPPH activity was strong. All the samples with herbs had lower residual nitrite levels compared to the samples with curing salt. Borage had a stronger influence on colour and antioxidant stability of the meat samples compared to hyssop. However, both herbs can be used as colour-forming and antioxidant agents along with nitrate-reducing bacteria.

## 1. Introduction

Avoiding synthetic additives with a preference towards natural ingredients is a trend that is observed on the food market [1], and processors are constantly trying to follow these demands. However, there is still no good alternative to the use of synthetic nitrites or nitrates in meat processing. Nitrites/nitrates are a group of compounds that not only have the function of a preservative, but also influence flavour and colour while inhibiting oxidation processes [2]. 

One of the problems associated with the potential elimination of nitrites/nitrates from meat processing is contamination with *Clostridium botulinum*. It is recommended that, in products without nitrites or with a low nitrite concentration, factors such as low pH and the addition of sodium chloride, antimicrobials should be incorporated to control microorganisms such as *C. botulinum* and *L. monocytogenes* [3].

The number of recent publications with various concepts shows that replacing nitrates is still a relevant and important issue. Chitosan with lycopene or acid whey were tested and the authors suggested that they were effective colour-forming and antioxidant agents [4,5]. A trial using the naturally formed zinc protoporphyrin was also conducted [6]. A different approach was presented by Jung et al. [7] or Lee et al. [8], who have proven that the atmospheric plasma (in which the nitrogen product from the atmospheric nitrogen was created) can be a good curing agent. Plants which are rich sources of nitrates (celery, spinach, lettuce, beet root, seaweed extract in their fresh, dried or fermented form), were used [5,9,10,11,12]. Another approach presented in these studies is the use of borage and hyssop as meat curing agents. *Borago officinallis* is cultivated for medicinal and culinary purposes. Among other wild plants, borage is popular for consumption in Italy and many other countries. Borage seed oil is common at pharmacies in the form of capsules to be injested as a dietary supplement. In studies on rats, it has been shown that it is effective in the treatment of diabetes [13], and because of its gamma-linolenic acid content, it may have effects in treating rheumatoid arthritis [14]. The tests conducted on *Artemia salina* did not show any toxicity of borage [13]. In tests conducted on rats, it has been indicated that borage consumption may lead to improvement in AD-induced cognitive dysfunction [15]. The plant’s green parts or flowers are known for their anti-inflammatory effects [16]. In naturopathic practice, borage is used for metabolism and hormone regulation [17]. Gilani et al. [18] stated that traditional borage extracts are used to help in hyperactive gastrointestinal, respiratory and cardiovascular disorders. Various parts of this plant contain different chemical constituents; therefore, it can be utilised in many ways. As borage is grown mainly for seeds and oil extraction; the rest of the plant is often treated as waste material and sold in the form of herbal teas [19]. 

Hyssop (*Hyssopus officinallis*) belongs to the *Lamiaceae* family and is rich in volatile, aromatic compounds. Its essential oil contains high amounts of *iso*-pinocamphone, *cis*-pinocamphone, *β*-pinene, terpinen-4-ol, pinocarvone, caracerol, *p*-cymene, elemol and myrtenal. It is widely used in the food or pharmaceutical industry [20]. Depending on the essential oil composition—which is also territory dependent—it may have variable antimicrobial effects [21,22,23]. Traditionally, it has been used to treat respiratory diseases or to improve appetite and digestion. An anti-inflammatory role in the asthmatic mouse model was confirmed by Ma et al. [24], along with the herb’s regulatory function in the immune system. Moreover, hyssop decreases the level of blood sugar and helps in muscle relaxing activity (tested on isolated intestinal preparations), probably due to the presence of iso-pinocamphone [25,26]. *Hyssopus officinalis* and *Borago officinalis* were chosen because of their traditional nature and the general positive opinion about herbs as food additives. These plants (similarly to other leafy vegetables) have a natural ability to accumulate nitrates [27,28]. Therefore, it was assumed that their effects on meat products could be tested. The addition of herbs to meat products may be beneficial for consumers in terms of the herbs’ positive medicinal properties. Moreover, they could be an alternative for consumers, who tend to avoid nitrites in meat products.

## 2. Materials and Methods

### 2.1. Experiment 1. Evaluation of Herbs

In the first stage of our research, we analysed the dried herbs of *Borago officinalis* and *Hyssopus officinalis* obtained from different sources (3 samples from each source) (water, ash, protein, fat, carbohydrate, polyphenol, nitrates and nitrites content). 

#### 2.1.1. Herb Analysis

Herbs were powdered using a grinder (Retsch, Grindomix GM200, Haan, Germany) and were analysed for water, protein, fat, ash and carbohydrate content according to recommendations of AOAC (Association of Official Analytical Chemists) [29]. All the analyses were conducted in triplicate.

##### Nitrite and Nitrate Content in Herb Samples

The analyses were conducted based on PN -92 A75112 [30]. An amount of 10 g of the powdered herb (Unidrive × 1000, CAT Scientific, Ballrechten-Dottingen, Germany) sample and 1 g of active carbon were weighed into 200 mL volumetric flask and rinsed with water. An amount of 5 mL of a sodium tetraboratedecahydrate saturated solution was added and the whole mixture was heated for 15 min. in a water bath (100 °C). After that time, the solution was cooled and 1 mL of Carrez I solution (potassium hexacyanoferrate II trihydrate and 1 mL of Carrez II (zinc acetate solution; Zn(CH_3_COO)_2_·2H_2_O) solution were added, mixed and left for 30 min. Then, the solution was filled up to 100 mL with distilled water, mixed and filtered through a filter paper. A Griess reaction solution consisting of Griess I (0.40 g of sulfonamide in 20 mL hydrochloric acid and 180 mL water), and Griess II (0.10 g *N-*(1-naphtyl) ethylene diamine dihydrochloride in 100 mL water) were prepared fresh before the analyses in proportions of 1:1. An amount of 10 mL of the filtered solution was mixed with 10 mL of the Griess reaction solution and left for 20 min. Absorbance (λ = 520 nm) was measured using spectrophotometer (Helios γ07-056, Thermo Scientific, Waltham, MA, USA). The concentration of nitrites was measured using the previously prepared standard curve. All the analyses were conducted in triplicate. 

For the nitrate analysis, 20 mL of the solution previously prepared for the nitrite analysis was measured into a 50-mL flask with 4 g of cadmium. The flask was closed and shaken for 15 min. Then, the solution was filtered through Whatman No. 1 filter paper into a 100 mL volumetric flask and filled up with distilled water. Ten mL of the filtered solution were mixed with 10 mL of the Griess solution and left for 20 min. The absorbance (λ = 520 nm) was measured using a spectrophotometer (Helios γ07-056, Thermo Scientific, Waltham, MA, USA). The concentration of nitrite was measured using the standard curve prepared beforehand. All the analyses were conducted in triplicate. 

### 2.2. Experiment 2. Application of Herbs to Meat

#### 2.2.1. Meat Sample Preparation

A pork shoulder obtained from a local retailer was minced (MADO MEW 613, Dornhan, Germany) and mixed manually with other ingredients as presented in Table 1. Three independent batches (every batch weighting 1 kg) were prepared, with each replication corresponding to a different manufacturing day. For each batch, a separate pork shoulder was used.

The herbs were powdered (Retsch, Grindomix GM200, Haan, Germany), bacterial culture Texel Meat Cultures NatuRed LT (*Staphylococcus carnosus*) (Danisco, Poznan, Poland) was mixed with water. Curing salt (NaCl 99.4%; NaNO_2_ 0.6%) was used. All the ingredients were mixed manually with meat (5 min). Meat samples contained: curing salt (CS), hyssop and salt (H), hyssop, salt and nitrate-reducing bacterial culture (HB), borage and salt (B), borage, salt and nitrate reducing bacteria (BB), salt and nitrate reducing bacteria (SB). Meat batters were placed in plastic test tubes and closed with plastic corks. They were incubated in a water bath (40 °C/90 min as recommended by the bacterial culture producer) (DKZ-3, Chemland, Poland), cooked (90 °C/20 min), chilled at room temperature and stored in refrigerated conditions (4 °C). The tests were conducted on the 1st, 8th and 15th days following the production. 

#### 2.2.2. TBARS (Tiobarbituric Acid Reactive Substances)

The analysis was conducted according to Pikul et al. [31] with some modifications. The sample (10 g) was homogenised (Unidrive × 1000, CAT Scientific, Ballrechten-Dottingen, Germany) with 34.25 mL of a cold (4 °C) extracting solution (4% perchloric acid and 0.75 mL of BHT (butylated hydroxtoluene) in ethanol) and 1 g of active carbon to remove the green colour of the sample. It was filtered through Whatman No. 1 filter paper into a 50-mL metric flask and adjusted to 50 mL using 4% perchloric acid. An amount of 5 mL of the filtrate was transferred to a 20-mL probe containing 5 mL of a TBA water solution (0.02M). The probe was closed and heated in a hot water bath (90°C) (DKZ-3, Chemland, Stargard Szczecinski, Poland) for 1 h, then cooled to room temperature. Absorbance was determined using a spectrophotometer (Helios γ07-056, Thermo Scientific, Waltham, MA, USA) at 532 nm against a blank containing 5 mL of 4% perchloric acid and 5 mL of the TBA reagent. TBARS values were expressed as mg of malonaldehyde in 1 kg of the sample and calculated by multiplying the absorbance values by the K coefficient of 5.5. The analyses were performed in duplicate.

#### 2.2.3. Measurements of the Antioxidant Capacity 

The meat extracts were prepared by homogenising 5 g of the sample (Unidrive × 1000, CAT Scientific, Germany) with 40 mL of ethanol (95%). The homogenates were vortexed in an ultrasonic bath (Sonic-6, Polsonic, Warsaw, Poland) for 15 min and then centrifuged (MPW-350, MPW, Warsaw, Poland) for 20 min at 3200× *g*. The extracts were filtered through Whatman No.1 filter paper into 50-mL flasks and filled with ethanol (95%). 

DPPH (2,2-diphenyl-1-picrylhydrazyl) radical scavenging activity of the meat samples was determined as described by Wu et al. [32]. A volume of 1.5 mL of each sample was added to 1.5 mL of 0.1 mM DPPH in 95% ethanol, mixed and left for 30 min at room temperature. After that time, absorbance was measured at 517 nm using a spectrophotometer. The antioxidant effect was expressed as:((Blank absorbance − Sample absorbance)/Blank absorbance) × 100%.

The experiment was carried out in duplicate.

FRAP (ferric reducing antioxidant power) analysis was conducted using the method described by Benzie and Strain [33]. Briefly, 0.4 mL of the meat extracts were mixed with 3.6 mL of the freshly prepared ferric-tripyridyltriazine (TPTZ) reagent (300 mM acetate buffer (pH 3.6), 8 mmol 2,4,6-tris(2-pyridyl)-*s*-triazine in 30 mM/l HCl; 20 mM FeCl_3_ in the ratio of 10:1:1). The mixtures were incubated at 37 °C for 10 min. and the absorbance of the TPTZ complex (formed with the reduced ferrous ions) was read at 593 nm against a blank sample using a spectrophotometer. The analyses were carried out in duplicate. The results were calculated from a standard scale of FeSO_4_·7H_2_O and expressed as mM Fe^2+^/L. 

#### 2.2.4. Colour Parameters

The colour of all the samples was measured using Konica Minolta CM-3500d (Tokyo, Japan) spectrophotometer. The CIE L*, a*, and b* values, were determined from the mean of 6 random readings on the cut surface of each sample. The target mask of an 8-mm area, CIE illuminant D65 and 10-degree standard observer angle were used. The instrument was calibrated on a black glass, then a white enamel tile following the manufacturer’s specifications. 

#### 2.2.5. Nitrites and Nitrate Content in Meat Samples

The analyses were conducted based on the PN-74/A-82114 procedure [34]. An amount of 10 g of the powdered herb (Unidrive × 1000, CAT Scientific, Ballrechten-Dottingen Germany) sample and 1 g of active carbon were weighed into a 200-mL volumetric flask and rinsed with water. An amount of 5 mL of a sodium tetraboratedecahydrate saturated solution were added and the whole mixture was heated for 15 min. in a water bath (100 °C). After that time, the solution was cooled and 1 mL of Carrez I (potassium hexacyanoferrate II trihydrate and 1 mL of Carrez II (zinc acetate; Zn(CH_3_COO)_2_·2H_2_O) was added, mixed and left for 30 min. Then, the solution was filled up to 100 mL with distilled water, mixed and filtered through a filter paper. The Griess reaction solution consisting of Griess I (0.40 g of sulfonamide in 20 mL hydrochloric acid and 180 mL water), and Griess II (0.10 g *N*-(1-naphtyl) ethylene diamine dihydrochloride in 100 mL water) were prepared fresh before the analyses in proportions of 1:1. An amount of 10 mL of the filtered solution were mixed with 10 mL of the Griess reaction solution and left for 20 min. The absorbance (λ = 520 nm) was measured using a spectrophotometer (Helios γ07-056, Thermo Scientific, Waltham, MA, USA). The concentration of nitrites was measured using the previously prepared standard curve. The nitrites content was recalculated for 1 kg of the sample. All the analyses were conducted in duplicate. 

For nitrate analysis, 20 mL of the solution previously prepared for the nitrite analysis was measured into a 50-mL flask with 4 g of cadmium. The flask was closed and shaken for 15 min. Then, the solution was filtered through a Whatman No. 1 filter paper into a 100-mL volumetric flask and filled up with distilled water. Ten ml of the filtered solution were mixed with 10 mL of Griess reaction solution and left for 20 min. The absorbance (λ = 520 nm) was measured using a spectrophotometer. The concentration of nitrites was measured using the standard curve prepared beforehand. The nitrate content was recalculated for 1 kg of the sample. All the analyses were conducted in duplicate. 

### 2.3. Polyphenol Characterisation by HPLC

The samples were prepared according to Klimczak et al. [35], with some modifications. The herbs (0.1 g) were powdered (Retsch, Grindomix GM200, Haan, Germany) an and the absorbance of the TPTZ and the absorbance of the TPTZ (1 g) were homogenised (Unidrive × 1000, CAT Scientific, Ballrechten-Dottingen, Germany), 1% L-ascorbic acid in methanol was added and mixed in a vortex (Laabnet, Edison, Edison, NJ, USA) and sonified for 15 min at 20 °C in an ultrasound mixer (InterSonic, Olsztyn, Poland). After that, the solution was mixed (v:v, 1:1) with NaOH (2 M). It was vortexed again (Labnet, Edison, NJ, USA) and kept in the dark for 4 h at room temperature. The pH of the samples was adjusted to 2.1–2.6 with HCl (2 M) (pH-meter Metrohm, Herisau, Switzerland). Then, the samples were centrifuged (MPW-352 RH, Rokow, Poland) (10 min, 4 °C, 1600× *g*) and transferred into a 5 mL volumetric flask with 1% L-ascorbic acid in methanol. Before HPLC analysis, the samples were centrifuged (MPW-260R, Rokow, Poland) (18 min., 4 °C, 18,000× *g*.) and filtered using PTFE-L filters (pore diameter of 22 µm).

Analysis was conducted using an HPLC set (Dionex UltiMate 3000) with a DAD detector (Thermo Scientific, Dreieich, Germany) on a Cosmosil 5C_18_-MS—II 250 × 4.6 mm ID, 5-µm column (Nacalai Tesque INC., Kyoto, Japan). The mobile phase consisted of two eluents: A—2% acidic water solution and B—100% methanol, flow 1 mL/min. The analysis was set for 50 min: eluent A—10 min 70%; 25 min 50%; 35 min 30%; 40 min 95%; 50 min 95%.

### 2.4. Statistical Analysis

All the herbs from various producers were analysed in triplicates (proximate analysis, nitrate/nitrite content) and quadruplicates (polyphenol profile). The data from experiments including herbs were subjected to one-way analysis of variance for each herb separately. The Tukey test (*p* < 0.05) was used to detect the differences between herbs of the same kind but from various producers. One-way analysis of variance was also performed for the results of polyphenol content in meat samples during storage. 

There were 3 independent production batches manufactured on different days. The differences between treatments (curing salt (CS), hyssop and salt (H), hyssop, salt and nitrate-reducing bacterial culture (HB), borage and salt (B), borage, salt and nitrate reducing bacteria (BB), salt and nitrate reducing bacteria (SB)) stored and analysed on the 1st, 8th and 15th day after production were analysed using two-way repeated measures ANOVA with two variables: treatment and storage time. All data were presented as mean values ± standard errors. The Tukey test at a significance level of 0.05 was used to locate significant differences between the means. Pearson’s correlation coefficient was calculated for the meat samples. STATISTICA 13 (TIBCO, Palo Alto, CA, USA) software was used for the analysis. 

## 3. Results 

### 3.1. Proximate Composition

In the first stage of the experiment, commercially available herb mixtures of *Hyssopus officinalis* and *Borago officinalis* (four of each kind) were tested to be selected for further analysis. Based on the results of the nitrite/nitrate analyses (Table 2), hyssop H1 and borage B3 were chosen. 

### 3.2. TBARS, DPPH, FRAP

The results of antioxidant capacity measurements for meat samples are presented in Table 3. There were significant differences in TBARS and DPPH values between treatments and during storage time. The interactions between the two variables were also significant. Thiobarbituric acid substances were cumulated in all the samples during storage but significant differences were noted only for the CS and BB samples. DPPH values were significantly higher in the samples containing herbs. DPPH scavenging activity decreased in all the samples after 15 days of storage. There was no noted effect of storage time or interactions between treatments and storage time on FRAP values. The samples with herbs had significantly higher FRAP activity compared to CS and SB. 

### 3.3. Colour Parameters

The colour parameters measured in all the samples at the three stages of refrigerated storage are presented in Table 4. The statistical analysis showed significant effect of treatment, storage time and interactions between those two variables. On the first day of the analysis, the redness (a*) was significantly higher in the control sample but decreased after the 7-day storage and did not change from the 8th to the 15th day. In the other samples, the most effective, in terms of creating red colour, was borage with the bacterial culture. However, both BB and HB were comparable with the control sample. It is worth indicating that the redness of HB and BB was increased strictly because of the nitrite reactions as no other colorant was added. The lightness (L*) was stable in all the samples throughout the whole storage period. SB samples were significantly lighter than all the other meat samples containing borage or hyssop.

### 3.4. Nitrite and Nitrate Content

The nitrite and nitrate content in meat samples is presented in Table 5. The effect of treatment and storage time as well as the interactions between those two variables were statistically significant. The control sample, to which curing salt was added, clearly did not contain nitrates. There were considerable amounts of nitrites which became slightly lower during storage. Samples with herbs contained small amounts of nitrites and much higher amounts of nitrates. 

### 3.5. Phenolic Profile

The phenolic profile of herbs and meat samples is presented in Table 6, respectively. Because of the small amount of herbs used in each meat sample, the final concentration of polyphenols was also low. They did not appear in the samples without herbs, which was expected. 

There were significant differences between hyssop and borage in the polyphenol composition. There was over 10 times more *t*-cynnamic acid, 4 times more chlorogenic acid, 7 times more *p*-coumaric acid, over 30 times more ferrulic acid, 3 times more caffeic acid and 3 times more rutin in hyssop, while the average amount of hippuric acid was comparable in both herbs. No salicylic acid was detected in the hyssop and no quercetin appeared in borage. 

## 4. Discussion

The amount of nitrates in the herbs was the most important in terms of meat curing. The aim was to apply such an amount of herbs so as to maintain the legal requirements of EU regulations 1333/2008 [36] (with amendments), with regard to nitrites/nitrates used in the production process. Commercially available herbs are a dry mixture of leaves, flowers and stalks. Usually, there is no information on the time of the harvest. The results presented in Table 2 confirm the significant differences between commercially available herb mixtures, which may depend on the proportions of leaves, flowers and stalks [37,38,39,40]. Variations in the quality and chemical component content were also reported by Hajdari et al. [41] who analysed hyssop samples cultivated in various regions. If we consider that the amounts of spices used in production or culinary practice, which usually do not exceed 2%, the differences in protein or fat content are negligible. However, nitrate content must be taken into account. To be able to use herbs in commercial standardised production, the herbs cultivation would need to be carefully planned and controlled.

TBARS analysis is a well-established test conducted on meat products. It allows to detect malondialdehyde—the main product of fat oxidation but also others—giving colour reaction with thiobarbituric acid. Malondialdehyde is described as a toxic substance. The oxidation of meat products and the creation of unpleasant off-flavours is one of the reasons to use antioxidants in the production process. Controlling the level of malondialdehyde is essential not only from a sensory or technological perspective, but most importantly, because of consumers’ health [42]. In this experiment, the question was whether the amount of herbs that was sufficient to cure meat, but not exceeding the amount of nitrates or nitrites, would be enough to inhibit the oxidation process.

TBARS values were comparable in all the samples on the 1st or the 8th day of analysis (Table 3). Significant differences appeared after 14 days of storage. BB had significantly lower TBARS values than all the other samples. The highest TBARS results were noted for SB and CS. HB, H and B were comparable, which could be evidence that the herbs’ constituents to inhibit the oxidation processes to some extent. There is evidence that borage seeds aqueous extracts decrease the lipid oxidation and are good chelating agents [43]. The antioxidant activity of the borage meal extract was also demonstrated by Wettasinghe et al. [44] in a model meat system. Hyssop, on the other hand, was reported to have strong antioxidant properties [41,45] or weak ones according to other authors [20].

The presence of phenolic compounds could be responsible for these results. However, when comparing the polyphenol profile of the analysed herbs, hyssop contained much more polyphenols than borage and the samples containing hyssop were comparable with the products with curing salt or salt and bacteria. In the samples with borage salicylic acid and rutin were detected. They were not found in the samples with hyssop. Those compounds and higher amounts of caffeic acid could be a reason for the borage’s stronger antioxidant activity. However, hyssop also inhibited oxidation processes to some extent, which could be due to the presence of t-cinnamic acid and quercetin, but also higher amounts of chlorogenic and ferrulic acids compared to borage. It has been recognised that phenolic acids are potent antioxidant substances. Depending on the chemical structure, polyphenols may have differing abilities to chelate Fe^2+^ ions. Unfortunately, the same polyphenols may also act as prooxidants after they have served as a reducing agent [46,47,48]. 

The term phenolic compound includes a range of substances typical for plants. Those substances contain an aromatic ring with one or more hydroxyl substituents [49]. As mentioned above, there were differences among the herbs of the same kind but coming from various sources, also, with regard to the polyphenol content (Table 6). In the research by Mhamdi et al. [50], the main phenolic acid detected in the borage seed was rosmarinic acid, which was not found in any of the analysed samples. There were also 15 mg of rosmarinic acid in the dried leaves of borage [51]. In other studies by Mhamdi et al. [52], in which the amount of polyphenols was analysed throughout stalk leaf development, syringic acid was predominant on the 60th day after the appearance of cotyledon and then sinapic and rosmarinic acids on the 105th day following cotyledon formation. The differences appearing among various herb samples (resulting from variation in isolation methods, collection time, locations or different chemotypes) may be the main cause of the inconsistencies presented in various studies [53]. 

There were no differences in polyphenol content between the meat samples containing borage (B and BB) or those including hyssop (H and HB) as the addition of the herbs was the same. The differences between samples containing borage and those containing hyssop result from differences in the polyphenolic composition of both herbs. More important was whether the discrepancies in polyphenol profile would influence the oxidation of those samples. After comparing these two parameters, it may be concluded that the polyphenol amount was inversely proportional to the antioxidant power. As mentioned above, the antioxidant effect may depend on the polyphenol profile. There was neither salicylic acid nor rutin in the hyssop samples and the borage samples contained an approximately doubled amount of caffeic acid, which may have affected the antioxidant power of borage samples. All of those compounds show an antioxidant effect [54,55,56], which was reflected in low TBARS values for all the samples containing borage (both with and without bacterial culture).

In research on sausages with fermented spinach or celery juice, the oxidation level was comparable with the control (nitrite cured) sample [12,57]. The results obtained in these studies (Table 3) clearly indicate very strong antioxidant properties of borage. After 15 days of storage the TBARS values in both B and BB samples were significantly lower compared to those with curing salt. This proves that in the products with herbs there were other antioxidant mechanisms than just NO. 

The phenolic content, which we expected to increase antioxidant capacity, might have been too low for the effects to be visible (only 0.5% used in the experiment). There are studies in which it was shown that despite the strong antioxidant activity of some herbs, they did not decrease the malonaldehyde content of a product [58]. Another explanation is that in a complicated matrix which was composed (meat, salt, nitrite reducing bacteria and herbs) the chemical reactions are hard to predict [59]. The antioxidant effect in the analysed samples could also be acquired because of the *Staphylococcus* strain used in the experiment. Analysing the results obtained here, it could be recommended to combine those two herbs in meat formulations to obtain higher antioxidant effects.

It has been underlined that oxidative stress caused by free radicals induces inflammation in the human body, which leads to the development of many diseases. Therefore, apart from monitoring the oxidation products, it was also important to know whether there is radical scavenging activity detected in the samples. DPPH is a commonly used method to measure the ability of a substance to scavenge free radicals. In the FRAP method, antioxidant power is measured by a redox reaction occurring between the substrate and Fe^3+^ ions, producing Fe^2+^ ions [60]. 

TBARS and DPPH values were inversely proportional (r = −0.64). The highest values of DPPH activity were noted for products with both herbs. What is interesting is that in the products with herbs and bacteria, the DPPH activity was lower compared to the products with only herbs. Moreover, it decreased during storage in all the samples. There is a possibility that the bacteria somehow reversed the antioxidant effect, which is significant in the context of the hypotheses. It was assumed that two effects could be gained: both that curing and the possible positive health effect (to be tested in the future research). It has been shown that in fermented sausages, dominated by *Staphylococcus* strains, the antioxidant activity is very strong [61]. However, according to the authors, this effect is obtained thanks to the presence of bioactive peptides and not the bacteria themselves. In this study, the microorganism activity was used only for the purpose of nitrate reduction, but according to Montel et al. [62], nitrate reductase activity is one of the reasons to use the *Staphylococcus* strains for oxidation prevention. The time of storage in our trial was quite short compared to the fermentation process used in sausage production. However, the antioxidant effect was reflected in TBARS analysis. 

All the samples showed very low ferric reducing activity (FRAP). DPPH activity values were the strongest during the first 7 days of storage and after another 7 days, they decreased significantly in all the analysed variants. Similarly to DPPH activity results in the samples with herbs, ferric reducing power was stronger in all the samples with herbs. Different mechanisms of oxidation prevention in the applied methods influenced the obtained results [63]. However, it may be concluded that in all the samples containing herbs, the antioxidant power was stronger compared to the control samples. Moreover, *Borago officinallis* seems to have stronger antioxidant stability than *Hyssopus officinallis*. 

Traditionally cured meat products are generally accepted by consumers. Therefore, it seems reasonable to compare experimentally cured samples to those with a standard amount of curing salt added. Colour measurement allows to evaluate the effectiveness of the curing method. Within the context of curing, the a* parameter expressing green-red components in the CIELab colour space, is one of the most important aspects. No less important is colour stability throughout the storage period [64]. The most effective in terms of redness was borage with bacterial culture. SB lightness values were comparable to the control samples, possibly because of the lack of any kind of green herbs. There were no statistically significant changes in b* value—neither among the variants nor during storage time. This was contrary to the results obtained by Horsch et al. [65] showing that hams with celery powder were more yellow. In fermented sausages with beetroot powder, the b* value was lower than in the control sample, however, at the end of the ripening period, all the samples were comparable [11]. The colour changes in various products cured with celery or beetroot powder depended on the type of meat used, the amount of the powder and on other additives like cherry powder or cranberry [11,57,66], therefore, it is difficult to compare the effectiveness of the applied methods directly. However, the a* values obtained in all the above studies were much higher compared to the results presented in this article. This quality trait may have an effect on sensory evaluation, especially taste. However, herbs, having their own specific taste, may mask the possible blandness of the meat samples. As was presented by Jin et al. [67], redness can be caused by red additives such as paprika or gardenia red, which may not change the flavour evaluation. 

The meat products monitored in several European countries contained from 0 to 150 mg/kg of nitrites, in some of them the maximum residual level set by EU was exceeded. The results obtained in the research by other authors are incomparable due to a number of factors affecting the results: the sampling procedure, the time after the production process, the actual level of salt added etc. Because of those multiple factors, it is impossible to detect the initial amount of the curing agent added at the production phase [68,69,70]. The levels of nitrates detected in our study were much higher than those detected in meat batters by Sucu and Turp [11]. Meat products usually contain fewer nitrites than nitrates as most of the substance added during the production process react with myoglobin, proteins, lipids and many other meat constituents [71]. It is considered beneficial to keep the residual nitrite level low because of the potential risk of nitrosamine formation [72]. Due to that, it is recommended to use reducing agents (e.g., ascorbic, erythorbic acid) along with nitrites [73]. On the other hand, the importance of nitrites dietary intake has been recently emphasized. Increased nitrate intake can be used in cardiovascular disease prevention, blood pressure regulation, protection against ischemia-reperfusion injury, inhibiting platelet aggregation, preserving or improving endothelial dysfunction. It can also enhance exercise performance [74,75]. The current established acceptable daily intake (ADI) for nitrate is 3.7 mg/kg of body mass (bm) per day [70]. In our research, the nitrates were not detected in the control sample, but were present in all the samples containing herbs, which, in light of the abovementioned statements, could be treated as a bonus. However, the constant concern to reduce nitrate and nitrite levels in processed meat makes this issue controversial. It is even more puzzling because the majority of nitrate intake comes from vegetables and not from food additives [70]. Such a concern may be caused by publications showing a negative impact of nitrite-cured meat on human health [76]. Those statements are repeated in the media although as it was stated by Swartz [77] “the association is not causation” and there is no direct evidence that psychological disorders are caused by nitrite cured meats. 

## 5. Conclusions

The differences between analysed herbs obtained from various sources show that using them in meat processing may make manufacturing standardisation difficult to obtain. The best results concerning colour are obtained when borage or hyssop are used along with bacterial cultures. The colour development and stability were comparable to the effects gained in the sample to which curing salt was added. The antioxidant effect was higher in samples containing herbs. However, it was not correlated with polyphenol content. 

Borage addition resulted in a higher antioxidant effect and colour development than in the case of hyssop. Further analyses concerning the inhibition of *Clostridium botulinum* development and sensory analysis should be conducted before testing on a semi-technical scale and promoting industrial recommendation.

## Figures and Tables

**Table 1 animals-10-02327-t001:** Composition of ingredients in production variants (% of meat amount).

	Treatment					
Ingredient	CS	SB	H	B	HB	BB
Curing salt	1.5					
NaCl		1.5	1.5	1.5	1.5	1.5
Bacterial culture		0.01			0.01	0.01
Hyssopus			0.5		0.5	
Borage				0.5		0.5
Water	6.7	6.7	6.7	6.7	6.7	6.7

Curing salt (CS), salt and nitrate reducing bacteria (SB), hyssop and salt (H), borage and salt (B), hyssop, salt and nitrate-reducing bacterial culture (HB), borage, salt and nitrate reducing bacteria (BB).

**Table 2 animals-10-02327-t002:** Proximate composition of herbs from different retailers [g/100 g] (mean values ± standard errors (SE).

Herbs	Water		Protein		Fat		Ash		Carbohydrate		NaNO_2_	NaNO_3_	
	Mean	SE	Mean	SE	Mean	SE	Mean	SE	Mean	SE	Mean	Mean	SE
**H1**	**11.41 ^a^**	**±0.07**	**15.45 ^b^**	**±0.22**	**1.67 ^c^**	**±0.03**	**8.58 ^c^**	**±0.32**	**71.47 ^b^**	**±0.26**	**0**	**1.81 ^b^**	**±0.007**
H2	10.22 ^a,b^	±0.19	17.56 ^a^	±0.06	2.00 ^b^	±0.09	11.87 ^b^	±0.17	70.23 ^b^	±0.14	0	0.24 ^d^	±0.002
H3	8.73 ^b^	±0.05	9.71 ^c^	±0.06	1.77 ^c^	±0.03	7.34 ^c^	±0.02	79.78 ^a^	±0.05	0	0.95 ^c^	±0.002
H4	12.27 ^a^	±0.79	17.58 ^a^	±0.19	2.53 ^a^	±0.03	14.78 ^a^	±0.37	67.62 ^c^	±0.77	0	4.32 ^a^	±0.140
B1	8.68 ^a^	±0.13	20.30 ^a^	±0.14	0.82 ^c^	±0.02	30.11 ^a^	±0.49	70.20 ^b^	±0.10	0	3.66 ^a^	±0.040
B2	9.13 ^a^	±0.07	15.40 ^b^	±0.10	1.32 ^b^	±0.01	13.08 ^c^	±0.44	74.15 ^a^	±0.18	0	1.32 ^d^	±0.040
**B3**	**7.99 ^a^**	**±0.02**	**20.94 ^c^**	**±0.06**	**1.47 ^b^**	**±0.02**	**17.51 ^b^**	**±0.10**	**69.60 ^b^**	**±0.06**	**0**	**1.88 ^b^**	**±0.050**
B4	9.25 ^a^	±1.13	18.55 ^d^	±0.12	2.27 ^a^	±0.04	9.98 ^d^	±0.03	69.94 ^b^	±1.07	0	1.46 ^c^	±0.040

^a,b,c,d—^Different letters in columns indicate significant differences between means separately for hyssop and for borage (*p* < 0.05). In bold: herbs chosen for further analysis.

**Table 3 animals-10-02327-t003:** TBARS, DPPH and FRAP values in samples during storage (mean values ± standard errors (SE).

Treatment	Storage Time (Days)	TBARS	(mg/kg)	DPPH	(%)	FRAP	(mmol Fe^2+^/L)
CS	1	0.94 ^cdef^	±0.45	25.87 ^cd^	±1.51	0.021 ^cde^	±0.001
	8	1.92 ^abc^	±0.08	15.54 ^def^	±0.94	0.045 ^bcde^	±0.001
	15	2.66 ^a^	±0.08	5.00 ^ef^	±0.84	0.031 ^e^	±0.004
SB	1	1.27 ^bcde^	±0.01	36.68 ^bc^	±2.70	0.018 ^cde^	±0.003
	8	1.72 ^ab^	±0.12	17.93 ^cde^	±3.25	0.065 ^abcde^	±0.010
	15	2.1 ^ab^	±0.13	7.24 ^f^	±2.22	0.036 ^de^	±0.012
B	1	0.95 ^def^	±0.02	72.59 ^a^	±1.29	0.059 ^bcde^	±0.003
	8	1.80 ^bcd^	±0.18	75.42 ^a^	±2.32	0.109 ^a^	±0.002
	15	1.36 ^cdef^	±0.24	49.25 ^bc^	±6.24	0.085 ^abcde^	±0.007
H	1	0.87 ^def^	±0.02	58.20 ^ab^	±2.10	0.041 ^bcde^	±0.001
	8	1.75 ^bcd^	±0.12	62.51 ^ab^	±2.42	0.095 ^abc^	±0.001
	15	1.93 ^a^	±0.35	33.14 ^cd^	±2.99	0.093 ^abcde^	±0.015
HB	1	0.29 ^f^	±0.02	55.01 ^ab^	±1.26	0.041 ^bcde^	±0.001
	8	1.20 ^def^	±0.22	60.06 ^ab^	±4.00	0.062 ^abcd^	±0.007
	15	1.86 ^abc^	±0.11	19.96 ^def^	±2.90	0.054 ^bcde^	±0.005
BB	1	0.24 ^f^	±0.05	64.83 ^ab^	±1.98	0.060 ^abcde^	±0.003
	8	1.28 ^ef^	±0.72	62.61 ^ab^	±1.60	0.092 ^ab^	±0.002
	15	0.92 ^def^	±0.09	28.02 ^cdef^	±4.74	0.080 ^bcde^	±0.013

^a–f^—different letters in columns indicate significant differences between mean values (*p* < 0.05). Curing salt (CS), salt and nitrate reducing bacteria (SB), borage and salt (B), hyssop and salt (H), hyssop, salt and nitrate-reducing bacterial culture (HB), borage, salt and nitrate reducing bacteria (BB).

**Table 4 animals-10-02327-t004:** Colour changes during storage (mean values ± standard errors (SE).

Treatment	Storage Time (Days)	L		a*		b*	
CS	1	56.58 ^bcde^	±0.42	7.03 ^a^	±0.41	10.03 ^a^	±0.11
	8	57.99 ^abcd^	±0.42	4.62 ^b^	±0.42	11.46 ^a^	±0.64
	15	58.38 ^abc^	±0.88	4.56 ^b^	±0.54	11.80 ^a^	±1.61
SB	1	58.28 ^abc^	±0.31	2.50 ^cde^	±0.60	12.60 ^a^	±0.21
	8	59.64 ^a^	±0.17	2.15 ^cde^	±0.34	13.26 ^a^	±0.89
	15	59.09 ^ab^	±0.55	2.28 ^cde^	±0.34	13.56 ^a^	±1.15
B	1	54.87 ^e^	±0.75	1.45 ^de^	±0.09	12.72 ^a^	±0.28
	8	55.45 ^cde^	±0.56	1.31 ^de^	±0.03	13.09 ^a^	±0.70
	15	55.42 ^cde^	±0.47	1.04 ^de^	±0.07	13.15 ^a^	±0.90
H	1	55.11 ^de^	±0.60	1.33 ^de^	±0.06	12.71 ^a^	±0.02
	8	55.09 ^de^	±0.78	1.29 ^de^	±0.06	12.79 ^a^	±0.55
	15	55.01 ^de^	±0.58	1.12 ^de^	±0.07	13.33 ^a^	±0.89
HB	1	54.50 ^e^	±0.74	3.06 ^bcd^	±0.56	10.50 ^a^	±0.47
	8	54.94 ^e^	±0.61	2.51 ^cde^	±0.26	11.99 ^a^	±1.04
	15	54.39 ^e^	±0.56	2.25 ^cde^	±0.49	11.93 ^a^	±0.89
BB	1	54.91 ^e^	±0.57	3.68 ^bc^	±0.52	10.57 ^a^	±0.57
	8	55.95 ^cde^	±0.35	2.85 ^bcde^	±0.51	11.81 ^a^	±0.74
	15	54.73 ^be^	±0.38	2.66 ^bcde^	±0.31	11.50 ^a^	±0.97

^a–e^—different letters in columns indicate significant differences between mean values (*p* < 0.05). Curing salt (CS), salt and nitrate reducing bacteria (SB), borage and salt (B), hyssop and salt (H), hyssop, salt and nitrate-reducing bacterial culture (HB), borage, salt and nitrate reducing bacteria (BB). lightness (L), green to red value (a*), blue to yellow value (b*)

**Table 5 animals-10-02327-t005:** Nitrites and nitrates content in meat samples (mg/kg) (mean values ± standard errors (SE)).

Treatment	Storage Time (Days)	Nitrites		Nitrates	
CS	1	100.38 ^a^ ±	10.05	0.00 ^e^ ±	0.00
	8	96.24 ^a^ ±	10.86	0.00 ^e^ ±	0.00
	15	71.25 ^a^ ±	20.46	0.00 ^e^ ±	0.00
SB	1	3.58 ^b^ ±	1.38	0.00 ^e^ ±	0.00
	8	2.32 ^b^ ±	0.60	0.00 ^e^ ±	0.00
	15	1.53 ^b^ ±	0.89	0.00 ^e^ ±	0.00
B	1	0.71 ^b^ ±	0.53	72.12 ^cd^ ±	7.03
	8	4.41 ^b^ ±	3.32	87.79 ^bcd^ ±	7.50
	15	1.81 ^b^ ±	1.21	72.84 ^cd^ ±	2.37
H	1	2.61 ^b^ ±	2.07	103.59 ^abc^ ±	4.40
	8	0.54 ^b^ ±	0.39	122.33 ^ab^ ±	16.01
	15	2.77 ^b^ ±	2.78	77.56 ^cd^ ±	2.15
HB	1	6.49 ^b^ ±	2.00	116.54 ^ab^ ±	7.59
	8	4.27 ^b^ ±	0.75	131.20 ^a^ ±	1.60
	15	5.77 ^b^ ±	1.87	102.12 ^abcd^ ±	17.12
BB	1	6.48 ^b^ ±	2.01	64.01 ^d^ ±	10.36
	8	5.22 ^b^ ±	1.91	89.82 ^bcd^ ±	8.61
	15	7.04 ^b^ ±	2.42	73.43 ^cd^ ±	1.31

^a–e^—different letters in columns indicate significant differences between mean values (*p* < 0.05). Curing salt (CS), salt and nitrate reducing bacteria (SB), borage and salt (B), hyssop and salt (H), hyssop, salt and nitrate-reducing bacterial culture (HB), borage, salt and nitrate reducing bacteria (BB).

**(a) animals-10-02327-t006-a:** 

Herb	t-Cynnamic	Quercetin	Chlorogenic Acid	Salicylic Acid	p-Coumaric Acid	Ferrulic Acid	Caffeic Acid	Rutin	Hippuric Acid
**H1**	**12.72 ^b^**	**±0.05**	**231.20 ^b^**	**±1.12**	**764.45 ^c^**	**±14.14**	**nd**		**23.94 ^d^**	**±0.25**	**92.65 ^d^**	**±1.09**	**31.70 ^d^**	**±0.20**	**nd**		**nd**	
H2	50.39 ^a^	±0.04	Nd		966.34 ^b^	±7.85	nd		64.57 ^b^	±1.25	668.65 ^a^	±7.58	433.85 ^a^	±6.61	128.59 ^b^	±2.14	85.14 ^a^	±1.99
H3	11.47 ^c^	±0.02	269.25 ^a^	±5.22	1412.98 ^a^	±5.10	nd		36.25 ^c^	±0.56	190.89 ^c^	±2.88	134.68 ^c^	±1.47	44.74 ^c^	±0.22	nd	
H4	50.39 ^a^	±0.08	44.41 ^c^	±1.40	1027.82 ^b^	±44.40	nd		72.50 ^a^	±0.59	417.45 ^d^	±4.77	348.55 ^b^	±6.08	143.66 ^a^	±2.04	60.22 ^b^	±2.43
B1	1.86 ^b^	±0.07	nd		139.96 ^b^	±2.57	52.63 ^d^	±1.15	3.68 ^b^	±0.03	nd		53.32 ^b^	±0.71	nd		88.05 ^a^	±1.85
B2	2.75 ^a^	±0.06	nd		407.03 ^a^	±8.71	167.13 ^a^	±1.04	10.61 ^a^	±0.02	14.26 ^a^	±0.12	132.63 ^a^	±2.47	52.79 ^a^	±0.36	nd	
**B3**	**2.53 ^a^**	**±0.02**	**nd**		**138.24 ^b^**	**±1.11**	**81.72 ^c^**	**±1.23**	**6.44 ^b^**	**±0.13**	**4.31 ^b^**	**±0.04**	**67.90 ^b^**	**±1.52**	**21.75 ^b^**	**±0.57**	**52.63 ^b^**	**±1.31**
B4	2.10 ^b^	±0.01	nd		209.73 ^b^	±2.38	94.34 ^b^	±1.60	5.47 ^b^	±0.15	nd		47.08 ^b^	±0.60	19.22 ^b^	±0.34	29.53 ^c^	±0.41

In bold: herbs chosen for further analysis; nd—not detected; ^a,b,c,d—^indicate significant differences between mean values ingroup of herbs of the same kind (hyssop or borage).

**(b) animals-10-02327-t006-b:** 

Variant	Time of Storage (d)	t-Cynnamic Acid	Quercetin	Chlorogenic Acid	Salicylic Acid	p-Coumaric Acid	Ferrulic Acid	Caffeic Acid	Rutin
CS	1																
8																
15																
SB	1																
8																
15																
B	1					7.20 ^1^	±0.22	5.93 ^1^	±0.23	0.36 ^1^	±0.04	0.53 ^1^	±0.02	3.06 ^1^	±0.33	1.10 ^1^	±0.04
8					5.29 ^1^	±0.63	6.14 ^1^	±0.17	0.37 ^1^	±0.03	0.84 ^1^	±0.06	3.36 ^1^	±0.40	1.18 ^1^	±0.06
15					3.83 ^2^	±0.49	5.84 ^1^	±0.19	0.31 ^1^	±0.02	0.73 ^1^	±0.01	2.89 ^1^	±0.27	0.90 ^2^	±0.06
H	1	0.51 ^1^	±0.04	5.73 ^1^	±0.17	27.89 ^1^	±0.91			0.83 ^1^	±0.10	2.49 ^1^	±0.35	1.51 ^1^	±0.18		
8	0.56 ^1^	±0.03	5.93 ^1^	±0.19	28.62 ^1^	±0.48			0.91 ^1^	±0.06	2.90 ^1^	±0.26	1.47 ^1^	±0.16		
15	0.51 ^1^	±0.04	5.41 ^1^	±0.33	26.75 ^1^	±1.29			0.89 ^1^	±0.03	2.88 ^1^	±0.32	1.38 ^1^	±0.14		
HB	1	0.47 ^2^	±0.03	4.87 ^2^	±0.25	23.50 ^2^	±0.87			0.73 ^2^	±0.09	2.26 ^1^	±0.33	1.44 ^1^	±0.09		
8	0.57 ^1^	±0.02	5.69 ^1^	±0.08	28.15 ^1^	±0.66			0.95 ^1^	±0.07	2.91 ^1^	±0.31	1.42 ^1^	±0.19		
15	0.46 ^2^	±0.02	4.56 ^2^	±0.33	23.72 ^2^	±0.99			0.76 ^2^	±0.01	2.55 ^1^	±0.11	1.37 ^1^	±0.11		
BB	1					7.56 ^1^	±0.48	6.15 ^2^	±0.12	0.36 ^1^	±0.04	0.55 ^1^	±0.07	3.42 ^1^	±0.30	1.17 ^1^	±0.07
8					5.98 ^1^	±0.53	5.99 ^2^	±0.17	0.39 ^1^	±0.04	0.69 ^1^	±0.05	3.50 ^1^	±0.39	1.17 ^1^	±0.04
15					5.42 ^1^	±0.56	6.88 ^1^	±0.18	0.42 ^1^	±0.01	0.66 ^1^	±0.03	3.23 ^1^	±0.76	1.19 ^1^	±0.05

^1,2^—different numbers indicate significant differences in the same sample analysed during storage (*p* < 0.05). Curing salt (CS), salt and nitrate reducing bacteria (SB), borage and salt (B), hyssop and salt (H), hyssop, salt and nitrate-reducing bacterial culture (HB), borage, salt and nitrate reducing bacteria (BB).

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
