# Peer review of "Potential Use of *Hyssopus officinalis* and *Borago officinalis* as Curing Ingredients in Pork Meat Formulations"

_animals, 2020, doi:10.3390/ani10122327_

Round 1
Reviewer 1 Report
Dear authors,
The study was designed, performed and reported very well. I have no comments to improve it except some minor corrections relating to language. Please see the attached file with highlights for that.
Regards,

Reviewer 2 Report
The paper entitled "The Potential use of Hyssopus officinalis and Borago officinalis as curing ingredients in pork meat formulations" by Marzena Zając et. al. deals with the effect of hyssop and borage as a curing substitute and its effect on meat in terms of antioxidant properties.
First of all, I would like to say that the design and the way the data has been reproduced is very aesthetic and the manuscript shows a high degree of professionalism.
To sum up, I have to say that I have to point out rather small formal errors and language and grammar mistakes. I have seen a few minor points in the result/discussion part which was not discussed further but which, in my opinion, has great potential for discussion. These are only suggestions on my part and are not mandatory, but I think these topics could add value to the manuscript.
The biggest problem I have is actually with the biological replicates. Although I am impressed by the statistical evaluation, I think that biological replicates of n=2 or n=3 are not sufficient to perform statistical evaluations.
As I find the manuscript otherwise very good, I am inclined to give the paper a minor and give the authors a chance to argue. The final decision whether this is justified is left to the journal editor and their decision.
As followed are my detailed comments
Comments
Line 101: It is best to add the chemical composition in brackets immediately after Griess I, afterwards Griess II, and the chemical composition in brackets. Writing the words twice in a row is confusing.
Line 103: Why didn`t you wrote the molecular formula? C12H16Cl2N2
Line 129: please remove the word “were prepared” at the end of the sentence.
Line 129: is “injected” the right word for it? How can I imagine that? Did you simply put the meat batters into a plastic test tube or did you actually inject them? Otherwise, I would rather suggest "were placed"
Line 131: please remove the “a” from “chilled at a room temperature”
Line 134: please introduce TBARS with its full name and the abbreviation in brackets afterwards in the headline
Line 135: I think it would be appropriate to briefly explain in an introductory sentence the importance of the TBARS analysis for people who are not experts please also in Line 146 for DPPH and FRAP. Otherwise, readers will feel left behind and may not read further.
Line 189: I would choose a different title. Twice just nitrites and nitrates is confusing and not enough. Please write explicitly nitrates content of the herbs or meat samples.
Line 189: please remove the point after Nitrates
Line 197: I have some minor problems with the section. There isn`t anything told about the amount of sample in the whole section. I would specify a quantity in Line 202 after the solution. With NaOH the unit, 2M/l is unusual. Please remove the litres.
Line 200: I think it should be called L-ascorbic acid
Line 204: Please also use M for the HCL concentration or %
Line 212: I think it should be called “A 2 % acidic water solution”?
Line 213: I think there is a number missing before the min. 95 %, maybe 45 min?
Line 214: I think the statistical methods are very good, but I doubt that you can perform statistical evaluations with n=2 or n=3. Maybe I get it wrong but if there are really only 2 or 3 measurements, I don't think that you can do a statistically relevant evaluation with it. If, for example, it was carried out on three experimental days and several samples were used on each day, it would be different. Then it should be mentioned that several samples were used per experimental day.
Line 261: the caption is a little bit confusing. Please introduce the abbreviations accordingly as they are presented in the table.
Line 261: It was left completely uncommented that borage with and without nitrate-reducing bacteria results in a reduction of the TBARS value during longer storage. This is an interesting result and should be mentioned explicitly. It seems that longer storage with Borage as a herb leads to a significant reduction of TBARS values and therefore has great antioxidant potential.
Line 263: analyzed should be changed to analysed because of BE
Line 282: there seems to be a double space line between “antioxidant mechanisms”.
Line 327: The "a" values are significantly lower compared to the CS. This means that there is much less reddening of the meat, which gives the meat its characteristic flavour. This also means that the meat appears paler and probably tastes blander. This should also be mentioned. As the inhibition of pathogens such as Clostridium plays a less important role in curing nowadays than the reddening of the meat and the characteristic aroma, this is of course an important factor. In fact, longer storage periods would certainly negate this effect, as with cured salts nitrites act directly and are decisive for reddening, whereas with HB and BB these have to be generated over time.
Line 355: If oxidation of nitrites to nitrate also occurs in cured salt meat, why are no nitrate concentrations measurable during the longer storage periods of CS? This, therefore, contradicts this statement
Line 365: Perhaps the addition of ascorbic acid during curing should also be mentioned, which leads to a low residual nitrite content and thus to lower nitrosamine formation
Line 381: The increased consumption of nitrite or nitrate-containing meat products is often criticised, e.g. increased likelihood of triggering manias (Chris Aiken: Mania Linked to Beef Jerky: Hot Dogs and Bacon May Be Next) or COPD (R. Varraso, R. Jiang et. al.: Prospective study of cured meats consumption and risk of chronic obstructive pulmonary disease in men. In American journal of epidemiology.) through the developing RNS.
Line 382: As borage has a much better antioxidant effect than hyssop, it could be concluded that this is due to the different composition of the phenolic compounds. In the comparison in Table 6, this could indicate that under certain circumstances certain phenolic components are irrelevant for the antioxidative effect. This would mean that caffeic acid and rutin would come into focus in terms of antioxidant activity. This should be taken into account in the discussion. Best supported by possible references to the effects of caffeic acid and rutin.
Line 420: please change standardization to standardisation due to BE
Reviewer 3 Report
Statements with a citation should fairly represent what is stated in the citation. For example, specify the species studied. Of note was one citation that concluded there could be potential harmful effects, but was listed as supporting positive effects of borage.
Statistical analysis must be improved. It appears from the methods that a factorial treatment design was used, with curing agent as the main effect and storage time as repeated measures. The analysis should reflect this design, and include tests for interactions between curing agent and storage time.

Round 2
Reviewer 3 Report
Number of samples, pooling of samples and resampling from the pool are described in the response to one of the reviewers, but still not clear in the manuscript.
One use of the word chemicals remains, although all herbs are chemicals.
